# Causes for Frequent Pathogenic *BRCA1* Variants Include Low Penetrance in Fertile Ages, Recurrent De-Novo Mutations and Genetic Drift

**DOI:** 10.3390/cancers11020132

**Published:** 2019-01-23

**Authors:** Pål Møller, Mev Dominguez-Valentin, Einar Andreas Rødland, Eivind Hovig

**Affiliations:** 1Department of Tumor Biology, Institute of Cancer Research, The Norwegian Radium Hospital, Part of Oslo University Hospital, Oslo 0424, Norway; Mev.Dominguez.Valentin@rr-research.no (M.D.-V.); einarro@ulrik.uio.no (E.A.R.); ehovig@ifi.uio.no (E.H.); 2Center for Hereditary Tumors, HELIOS-Klinikum Wuppertal, University of Witten-Herdecke, Wuppertal 42283, Germany; 3Department of Medical Genetics, The Norwegian Radium Hospital, Oslo University Hospital, Oslo 0424, Norway; 4Center for Bioinformatics, Department of Informatics, University of Oslo, PO box 1080, Blindern 0316 OSLO, Norway

**Keywords:** pathogenic *BRCA1* variants, founder variants, fitness, genetic epidemiology, genetic drift, hot-spots germline mutations.

## Abstract

*Background:* We have previously demonstrated that the Norwegian frequent pathogenic *BRCA1 (path_BRCA1*) variants are caused by genetic drift and recurrent de-novo mutations. We here examined the penetrance of frequent *path_BRCA1* variants in fertile ages as a surrogate marker for fitness. *Material and methods:* We conducted an observational prospective study of penetrance for cancer in Norwegian female carriers of frequent *path_BRCA1* variants, and compared our observed results to penetrance of infrequent *path_BRCA1* variants and to average penetrance of *path_BRCA1* variants reported by others. *Results:* The cumulative risk for breast cancer at 45 years in carriers of frequent *path_BRCA1* variants was 20% (94% confidence interval 10–30%), compared to 35% (95% confidence interval 22–48%) in carriers of infrequent *path_BRCA1* variants (*p* = 0.02), and to the 35% (confidence interval 32–39%) average for *path_BRCA1* carriers reported by others (*p* = 0.0001). *Discussion and conclusion:* Carriers of the most frequent Norwegian *path_BRCA1* variants had low incidence of cancer in fertile ages, indicating a low selective disadvantage. This, together with the variant locations being hotspots for de novo mutations and subject to genetic drift, as previously described, may have caused their high prevalence today. Besides being of theoretical interest to explain the phenomenon that a few *path_BRCA1* variants are frequent, the later onset of breast cancer associated with the most frequent *path_BRCA1* variants may be of interest for carriers who have to decide if and when to select prophylactic mastectomy.

## 1. Introduction

Female carriers of pathogenic variants of the *BRCA1 (path_BRCA1*) gene have risks for breast and ovarian cancer. Early diagnosis and treatment have not been demonstrated to cure all those having breast cancer [1] nor those with ovarian cancer [2], for which reasons prophylactic bilateral mastectomy (PBM) and prophylactic salpingo-oophorectomy (PBSO) are accepted options. Precise estimates of risk by age for breast or ovarian cancer in the carriers of *path_BRCA1* variants are important when selecting if and when to undertake PBSO or PBM.

In a series of reports, we have previously described the Norwegian frequent *path_BRCA1* variants [3,4,5,6,7,8]. Some details relevant to the present report are mentioned below. Included in these previous reports is a description of the underlying genetic paradigms with references. This may be summarized as follows: 25–30 generations ago, the Bubonic plagues killed most of the Norwegian population, leaving about 200,000 survivors. They grouped in small geographical isolates, followed by a rapid population expansion by in-mating inside these isolates until three generations ago, when modern communications allowed contact between the separated populations. Because of the population collapse, Norway as an independent state no longer existed, and the sailing traditions from the Viking ages could not be continued. When warships equipped with cannons later were invented, the traditional Norwegian ships could not accommodate the cannons. Thus, the Norwegian population was for practical reason isolated from the rest of the world until the end of the Napoleon wars in 1815. At this time, the population had expanded and was starving, and about half of the population emigrated to USA/Canada to survive. In sum, 200,000 survivors of the Bubonic plagues by in-mating in small geographical isolates expanded to about 10 million today, when counting the emigrants to USA/Canada. Assuming a 0.3% prevalence of *path_BRCA1* variants, the 200,000 survivors of the Bubonic plagues should have 600 *path_BRCA1* alleles. The probability for each one of these to survive in a stable population until today should be about 5%, which should in this case amount to 30 different *path_BRCA1* alleles. Genetic drift due to population bottlenecks and expansion, combined with in-mating in isolated populations may substantially change the expectations described above, and also the relative prevalence of each of the surviving *path_BRCA1* alleles. When describing the epidemiology of the *path_BRCA1* variants in Norway, we attributed the variations to genetic drift and one or a few hotspots for de novo germline mutations.

In 1999, we initially reported that our prospective findings indicated that carriers of the most frequent *path_BRCA1* variants had lower incidence for infiltrating breast cancer than other *path_BRCA1* carriers [3]. We confirmed this ten years later with our updated series [4]. In 2013, we reported that the most frequent Polish *path_BRCA1* variant had the same lower incidence for breast cancer as the Norwegian frequent *path_BRCA1* variants, while the North-American less frequent *path_BRCA1* variants implied higher risk for infiltrating breast cancer, similar to the less frequent Norwegian *path_BRCA1* variants [5].

In 2007, we reported that 2.5% of our incident breast cancer cases had one of the frequent Norwegian *path_BRCA1* variants, while 23% of the incident ovarian cancers had one of these variants [6]. Bearing in mind that the population incidence of breast cancer is about 10 times the incidence of ovarian cancer, these findings indicated that the risks for breast and ovarian cancers were similar in carriers of the Norwegian founder *path_BRCA1* variants, but with later onset of ovarian cancer. This confirmed our results from segregation analyses in these families, in that lifetime risks for breast and ovarian cancer in the former generation were similar [7]. Before a significant number of breast cancer cases were cured through modern health care, young onset breast cancer was a common cause of death in the *path_BRCA1* carriers. This is probably related to why the gene was named a BReast CAncer gene and an early perception was that *path_BRCA1* was predominantly causing disease in young adult women. The high lifetime incidence for ovarian cancer has in recent years been illustrated through some early-onset breast cancers being cured, and in carriers having undertaken PBM.

We have previously demonstrated that the most frequent *path_BRCA1* variant in Norway are located in poly-A tract of the gene (chr17:41246515-41246559 ATAGGCGGACTCCCAGCACAGAAAAAAAGGTAGATCTGAATGCTG (normal reverse complement, poly-A sequence into which an extra A is inserted the into the poly-A part in the pathogenic allele)). This may theoretically indicate hot-spots for recurrent de novo mutations, and we have demonstrated that the most frequent *path_BRCA1* variant in our series occurs in different haplotypes world-wide. We interpret this to reflect that this part of the gene is a hot-spot for de novo mutations. We also find that all examined Norwegian carriers have the same haplotype, as would be expected as a result of local genetic drift [8]. 

We decided to examine cancer incidence in fertile ages for the frequent *path_BRCA1* variants in Norway based on the methods developed for estimating cancer incidences in the Prospective Lynch Syndrome Database (PLSD) [9]. We hypothesized that the frequent *path_BRCA1* variants should not have reduced fitness, and we in this study used penetrance in fertile ages as a surrogate marker for fitness. Until recent generations, most children were born by mothers aged less than 35 years, and it is well recognized that dominantly inherited disorders implying reduced fitness below this age are rare, and that the prevalence is maintained by de novo mutations. In addition, a recent report by [10] describing the average incidence of breast cancer in *path_BRCA1* carriers provided the opportunity to compare the incidence of breast cancer in carriers of our frequent *path_BRCA1* carriers with the average incidence from other countries.

## 2. Material and Methods

### 2.1. Patients and ethics

We have previously described in detail how we from 1988 onwards established an out-patient inherited cancer clinic, a combined electronical medical file and a research registry holding all information received from all patients. Likewise, we established a biobank with blood sample from the genetically tested patients including informed consent for research studies [6]. All genetic testing was performed according to national legislation for clinical use. All patients filed had either referred themselves or had been referred by other hospitals/physicians. All patients were followed with annual mammography and annual breast MRI for breast cancer, and with transvaginal ultrasound and blood CA 125 for ovarian cancer. The study was approved by the Ethical review board (No. S02030) and by The Norwegian Data Inspectorate (No. 2001/2988–2).

### 2.2. Material and Methods

For the present study, we updated our series and calculated cumulative incidence by age with a more precise method than that used in our previous reports as referenced above. These methods have been described in detail elsewhere [9]. In brief, the methods calculate annual incidence rate in five-year cohorts and then calculate cumulative risk by age based on these. Doing so, differences of cancer incidences by age, different numbers of patients in the different age groups, and inclusion of patients at different ages are adjusted, corresponding to how cancer registries calculate cumulative incidences. Age at first prospectively planned and executed examination was used as inclusion point for follow-up. 

When considering ovarian cancer incidence, all cases with oophorectomy for any reason before or at inclusion were excluded. The first ovarian cancer diagnosed after inclusion was scored as event, and any other cancer—including breast cancer—was ignored. Series were right-censored at oophorectomy for any reason or last examination, whichever came first.

When considering breast cancer incidence, all cases with breast cancer before or at inclusion were excluded, as were all cases with oophorectomy for any reason before or at inclusion. First breast cancer diagnosed after inclusion was scored as event. Besides ovarian cancer implying oophorectomy, any other cancer was ignored. Cases with mastectomy for any reason prior to or at the same age as inclusion were excluded, and series were right-censored if oophorectomy for any reason, mastectomy, breast cancer or last examination, whichever came first. Because of right-censoring at first breast cancer, this report contains no information on synchronous or meta-synchronous breast cancers.

According to national guidelines, follow-up was initiated from 25 years of age onwards, to the end that we have no information on annual incidence of cancer before that age, while we do have complete records of cancer before 25 years of ages used for the selection of patients to be studied as mentioned above. Annual incidence rates (AIR) in 5-year cohorts were calculated. Risk for cancer at age 25 years was set to zero as starting point for calculating cumulative incidences. Incidence rates of cancer past childbearing ages were not the foci for the present study, and because of high uptake of PBSO, the number of follow-up years was insufficient to calculate incidence rates in 5-year age cohorts after 50 years of age with the methods employed. Similarly, we did not have enough observation years to calculate annual incidence of breast cancer in fertile ages after oophorectomy, because survival after ovarian cancer has been very poor, and because PBSO before the end of childbearing ages has not been implemented. Previously, we have scored infiltrating cancer only as event, while we for this report decided to use the recent report by [10] as contrast group, and thus included carcinoma in situ as event, as they did. As a methodological control, we applied our algorithms as previously described [9], using the follow-up years and events given by [10], and came to similar cumulative risks in their series as they did with their methods.

Fishers’ exact test was used for comparisons of breast cancer incidence in the specified age cohorts. A detailed description of the methods used to consider the differences between distributions is included as on-line Appendix A.

## 3. Results

When the series were censored for the current study, all families meeting our criteria for being genetically tested had been examined for the 8 most frequent *path_BRCA1* variants [6]. Also, a large number of them had been examined for other Norwegian *path_BRCA1* variants or were fully *BRCA1* sequenced. In line with our previous reports, *BRCA1* c.1016insA, c.1556delA, c.3228delAG and c.697delGT were grouped as frequent, and the others as infrequent. Among 1918 demonstrated carriers of *path_BRCA1* variants, 1055 (55%) had one out of the four most frequent variants. 

Distribution of observation years, cancer cases and annual incidence rates (AIR) as events/observation years, starting at 25 years of age, is given in Table 1. No ovarian cancer was observed before 35 years of age. Cumulative risks for breast cancer as calculated from the annual incidence rates in the 5-year cohorts are given in Table 2, and are illustrated in Figure 1. The distribution of numbers with each single separate variant is illustrated in Figure 2 and where the four frequent variants are identified. Table 3 describes all *path_BRCA1* variants and their prevalence. As seen, the carriers of frequent *path_BRCA1* variants had lower cumulative incidence of breast cancer than carriers of infrequent *path_BRCA1* variants (*p* = 0.05 at age 40, *p* = 0.02 at age 45 years). The cumulative incidences for breast cancer for carriers of each of the four frequent *path_BRCA1* variants are illustrated in Figure 3: one carrier of *BRCA1c.3822delAG* developed breast cancer before 35 years of age, and none in the others. This was the only breast cancer detected before 35 years of age in carriers of any of the eight most frequent *path_BRCA1* variants. 

Compared to the average cumulative risk for *path_BRCA1* carriers elsewhere reported by [10], the carriers of Norwegian frequent *path_BRCA1* variant had significantly lower cumulative incidences (*p* = 0.0001 at ages 40 and 45 years).

As seen in supplementary eFigure1a in the report by [10]., they observed no breast cancer before 25 years of age, to the end that when we set the cumulative incidence in their series to zero for the comparisons applying our methods presented here, this implies no error. On the other hand, not having access to the exact number of observation years from 25–30 years in their series, we used their average annual incidence rate from 20–30 years indicated in their Table 3 when calculating cumulative incidence from 25–30, which resulted in a slight underestimation of the cumulative risk in their series (24% at age 40 calculated by us as seen in Table 2, compared to their reported 26% at age 40). This difference had no impact on the conclusions in this report: if we had calculated the significance for difference between their series using their point estimates and confidence intervals and our frequent *path_BRCA1* variants, the p-values given in Table 2 would have been lower.

Comparing the annual incidence rates for breast cancer in the age group 21 to 29 years, the point estimate for observed incidence rate in carriers of infrequent *path_BRCA1* variants was higher (13 events in 596 observation years; AIR 2.2% (95%CI 1.7–3.7%)) than in carriers of frequent *path_BRCA1* variants (8 events in 887 observation years; AIR 0.9% (95%CI 0.4–1.8%)) but insignificantly so (*p* > 0.05). Comparing the incidence in carriers of our frequent *path_BRCA1* variants with the corresponding annual incidence rate reported as average for *path_BRCA1* carriers by [10], the average cumulative incidence reported by Kuchenbaecker et al. was significantly higher (83 events in 3,348 observation years; AIR 2.5% (95% CI 2.0–3.1%)), than in our carriers of frequent *path_BRCA1* carriers (Fisher exact *p* = 0.03). 

No carrier of frequent *path_BRCA1* variants in our series had any cancer before or at inclusion, corresponding with the inclusion criteria used by [10].

## 4. Discussion

The most frequent *path_BRCA1* variants in our series had low penetrance in fertile ages, and the penetrance was lower in carriers of infrequent *path_BRCA1* variants in our series, and lower than the population average for *path_BRCA1* carriers reported by others. Infrequent *path_BRCA1* variants had penetrance in fertile ages similar to the population average for the *path_BRCA1* variants elsewhere reported. Our report is to our knowledge the first report focusing cancer incidence in fertile ages in prospectively detected carriers of *path_BRCA1* variants in order to consider why some *path_BRCA1* variants become more frequent than others.

For a de novo mutation to survive, the carrier has to produce children: to escape Darwinian de-selection, the carriers have to be healthy during fertile ages. This is why, in general, all dominantly inherited disorders with serious prognosis manifesting themselves in young adult ages are de-selected. Their prevalence in the population is a balance between de novo mutations and Darwinian de-selection.

The most frequent *path_BRCA1* variants in our series is one additional A in a poly-A part of the *BRCA1* gene, theoretically indicating increased probability for this pathogenic variant to be inserted into the population more than once. This variant is indeed reported from other countries to have occurred more than once, as indicated by the occurrence on multiple different *BRCA1* haplotypes, one of these being prevalent in the Norwegian population [8]. 

Previously, we have discussed that the probability for a random pathogenic variant to become frequent increases with inbreeding in expanding populations [11], which indeed is the history for the Norwegian population through the last 600 years. 

In summary, we now have described three independent factors which may all have contributed to make the most frequent Norwegian *path_BRCA1* variants frequent: Less de-selection because of lower risk for cancer in fertile ages, hot-spot for de novo mutations, and a high probability for genetic drift in areas of inbreeding in an expanding population.

Our results are derived from intense screening of *path_BRCA1* carriers aiming at early detection and cure to improve prognosis [1]. Breast cancer screening implies over-diagnosing [12]. If the series reported by [10]. had been subjected to less intense screening, the differences between their average results for *path_BRCA1* carriers and our series of frequent *path_BRCA1* carriers may be artificially low, which would strengthen the conclusions in this report. Reference [10] did not mention whether or not the first (prevalent) round cancers were included when calculating annual incidence rates. If the prevalent round findings were included in their series, their reported annual incidence rates would be artificially too high. This would be a confounder to the conclusions in this report, but this would have no impact for the internal comparison of frequent versus infrequent *path_BRCA1* variants in our series which was the original only focus for this report. We have performed one predetermined test with the same categorization of our series comparing the four most frequent variants versus infrequent variants keeping the same categorizations as in our previous reports: we have not done multiple testing in our material, which in case should have been corrected for when calculating the *p*-values.

In addition to inform on the mechanisms making some *path_BRCA1* variants frequent, our findings may be of interest for Norwegian carriers of *path_BRCA1* variants when considering if, and if so at which age, to undertake PBM. 

## 5. Conclusions

Considering the incidence of cancer in fertile ages as a surrogate marker of fertility/fitness, and based on the theory of Mendelian deselection when reduced fitness, we expected and confirmed that carriers of the most frequent Norwegian *path_BRCA1* variants had lower incidence of cancer in fertile ages than carriers of less frequent *path_BRCA1* variants. This adds to our previous reports demonstrating that the distribution of *path_BRCA1* variants are influenced by genetic drift and hot-spot for de novo mutations. Besides being of theoretical interest to explain why some *path_BRCA1* variants are more frequent than others, this is of interest to female variant carriers who are to decide if, and if so at which age, to undertake prophylactic mastectomy.

## Figures and Tables

**Figure 1 cancers-11-00132-f001:**
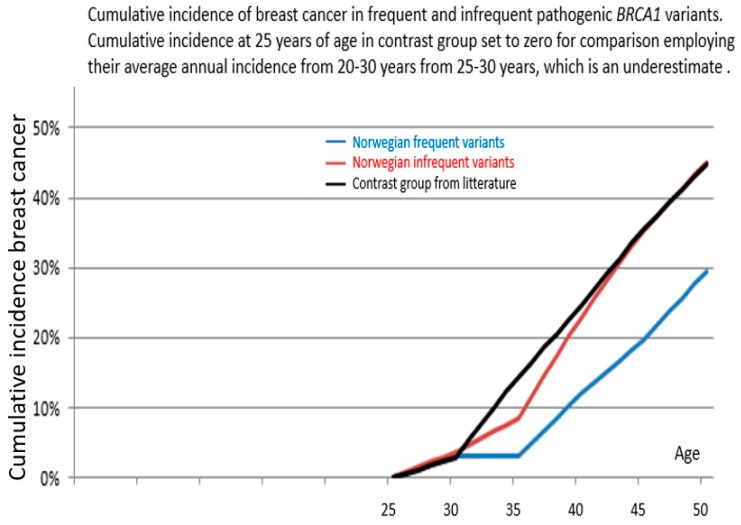
Breast cancer cumulative incidences in female carriers of frequent Norwegian *path_ BRCA1* variants (blue line), infrequent (red line) and contrast group (black line) starting from 25 years of age.

**Figure 2 cancers-11-00132-f002:**
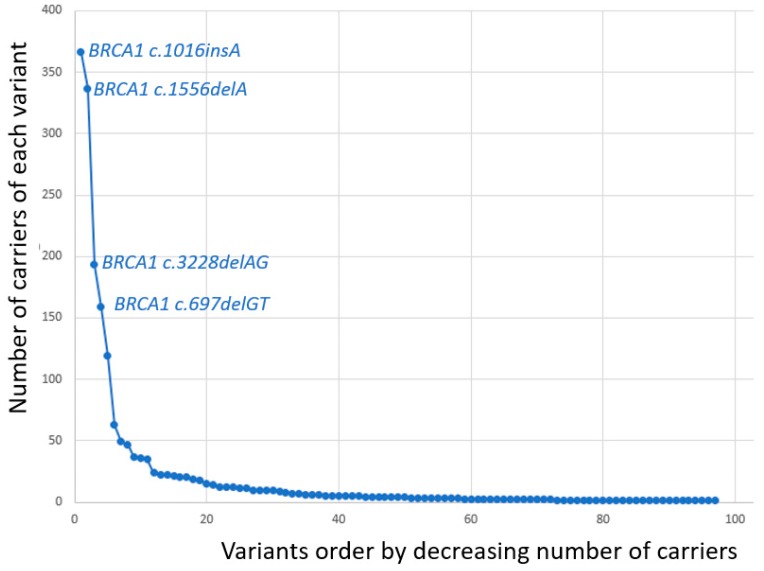
Number of carriers with each separate *path_BRCA1* variant demonstrated. The four variants in this and previous reports grouped as frequent are indicated.

**Figure 3 cancers-11-00132-f003:**
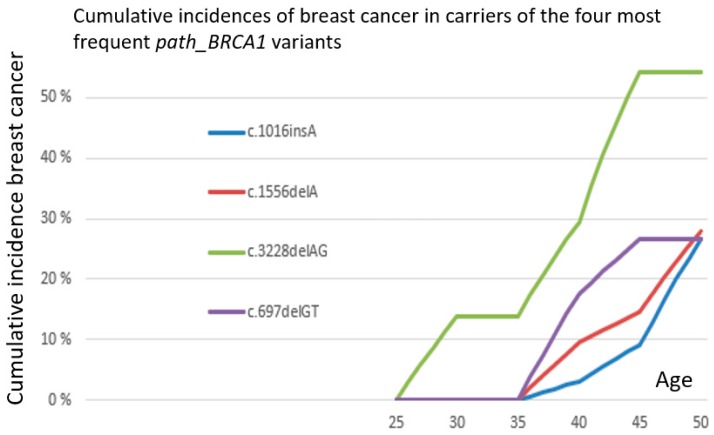
Separate cumulative incidences of breast cancer in carriers of the four most frequent *path_BRCA1* variants.

**Table 1 cancers-11-00132-t001:** Number observation years, number of events and annual incidence rates (events/observation years, AIR) in 25-year cohorts from 25 years of age onwards. 25yrs: observation years 25–29 years; 25ca: cancers observed 25–29 years; AIR25: annual incidence rate from 25 years of age included to 29 years included with 95% confidence intervals for breast cancer incidences; AIR30, AIR40 and AIR45 likewise.

Cancer Type	*Path_BRCA1* Variant	25yrs	25ca	AIR25 (95%CI)	30yrs	30ca	AIR30 (95%CI)	35yrs	35ca	AIR35 (95%CI)	40yrs	40ca	AIR40 (95%CI)	45yrs	45ca	AIR45 (95%CI)
Ovarian cancer	Frequent	48	0	0	161	0	0	173	1	0.0058	115	5	0.0435	33	1	0.0303
	Infrequent	36	0	0	100	0	0	107	0	0	80	1	0.0125	28	2	0.0714
Breast cancer	Frequent	165	1	0.006 (0–0.034)	463	0	0 (0–0.008)	414	8	0.019 (0.008–0.038)	219	4	0.018 (0.005–0.047)	80	2	0.025 (0.003–0.090)
	Infrequent	134	1	0.008 (0–0.041)	299	3	0.010 (0.002–0.029)	297	10	0.034 (0.016–0.062)	173	6	0.034 (0.013–0.076)	95	3	0.032 (0.006–0.093)

**Table 2 cancers-11-00132-t002:** Cumulative incidence from age 25 years set to zero up to given age, 95% confidence intervals and p-values for differences between cumulative incidence in carriers of frequent *path_BRCA1* variants in Norway compared to carriers of infrequent variants in Norway and compared to average reported by [10].

*Path_BRCA1* Variant	From 25 Years of Age to	Cumulative Incidence	95% Confidence Interval	*p*-Value Versus Infrequent	*p*-Value Versus Kuchenbaeker et al.
Frequent	40 years	12%	4–20%	0.05	0.0001
	45 years	20%	10–30%	0.02	0.0001
	50 years	29%	14–45% *	0.09 *	0.01 *
Infrequent	40 years	23%	12–34%		
	45 years	35%	22–48%		
	50 years	45%	30–60% *		
Kuchenbaeker et al.	40 years	24%	21–28%		
	45 years	35%	32–39%		
	50 years	45%	41–49%		

***** Wider confidence intervals and corresponding higher *p*-values in old ages in our series due to low numbers, because of high uptake prophylactic salpingo-oophorectomy (PBSO) past childbearing ages.

**Table 3 cancers-11-00132-t003:** *Path_BRCA1* variants detected.

*BRCA1* Variant	Number of Carriers
c.1016dupA	366
c.1556delA	337
c.3228_3229delAG	193
c.697_698delGT	159
c.3178G>T	119
c.4745delA	63
c.1A>C	49
c.2351_2357delCGTTACT	47
c.5075-2A>C	37
c.3084_3094delTAATAACATTA	36
c.5047G>T	35
c.(441 + 1_442–1)_(4357 + 1_4358–1)del	24
c.(4185 + 1_41861)_(4357 + 1_4358–1)dup	22
c.3607C>T	22
c.3048_3052dupTGAGA	21
c.5266dupC	20
c.3331_3334delCAAG	20
c.(5332 + 1_5333–1)_(5406 + 1_5407–1)del	19
c.1072delC	18
c.5511G>A	15
c.1450G>T	14
c.(80 + 1_81–1)_(4986 + 1_4987–1)del	12
c.5513T>G	12
c.66dupA	12
c.2475delC	11
c.3966delA	11
c.2869C>T	10
c.2591C>G	10
c.1058G>A	10
c.3319G>T	10
c.4065_4068delTCAA	9
c.5407-25T>A	8
c.1292dupT	7
c.2558ins356	7
c.3874delT	6
c.68_69delAG	6
c.5251C>T	6
c.5534C>A	5
c.1687C>T	5
c.(?_1–1)_(4357 + 1_4358–1)del	5
c.794_795delCT	5
c.5503C>T	5
c.3710delT	5
c.4035delA	4
c.2989_2990dupAA	4
c.4689C>G	4
c.3319G>T	4
c.5213G>A	4
c.4300_4301delA	4
c.115T>G	4
c.5153G>C	3
c.848T>A	3
c.339_361dup	3
c.4612C>T	3
c.(?_1–1)_(134 + 1_135–1)del	3
c.457_458ins21	3
c.3770_3771delAG	3
c.2869C>T	3
c.3700_3704delGTAAA	2
c.3477_3479delAAAinsC	2
c.3835_3835delG	2
c.2438dupG	2
c.2389G>T	2
c.1695dupG	2
c.65T>C	2
c.1287_1287delA	2
c.385_385delG	2
c.2681_2682delAA	2
c.4932_4933dupAA	2
c.4972_4972delA	2
c.(5074 + 1_5075–1)_(5592 + 1_?-1)del	2
c.(134 + 1_135–1)_(441 + 1_442–1)del	2
c.241C>T	1
c.5434C>G	1
c.3817C>T	1
c.2722G>T	1
(4185 + 1_41861)_(4357 + 1_4358–1)del	1
c.4883T>C	1
c.1059G>A	1
c.5123C>T	1
c.3937C>T	1
c.140G>T	1
c.1961dupA	1
c.3808T>G	1
c.(5193 + 1_5194–1)_(5592 + 1_?-1)del	1
c.514C>T	1
c.2185G>T	1
c.4689C>G	1
c.130T>A	1
c.4987-16T>G	1
c.1674_1674dupA	1
c.2745_2748delTCAA	1
c.5075A>C	1
c.(4675 + 1_4676–1)_(4986 + 1_4987–1)del	1
c.929delA	1
c.3005delA	1
c.5212G>A	1
Sum	1918

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
