# Peer review of "Causes for Frequent Pathogenic BRCA1 Variants Include Low Penetrance in Fertile Ages, Recurrent De-Novo Mutations and Genetic Drift"

_cancers, 2019, doi:10.3390/cancers11020132_

Reviewer 1 Report

In this paper Moller et al. examine the penetrance of BRCA1 pathogenic variants that are frequent in the Norwegian population.  Penetrance was assessed in fertile years as a surrogate for evolutionary fitness. They assessed penetrance using an observational prospective study of Norwegian female carriers. These results were compared with penetrance of other not so frequent pathogenic variants in BRCA1 in the Norwegian population and with the average penetrance reported in other breast cancer studies. The authors determine that penetrance of these frequent pathogenic variants was significantly lower than for carriers of infrequent pathogenic variants. They conclude that the relative high prevalence of some pathogenic BRCA1 variants in the Norwegian population is due to genetic drift (although not directly assessed in this paper) and the low selective disadvantage of carriers of frequent pathogenic variants.

This is a short and focused manuscript that reports interesting observation that may however have limited implications for other populations but illustrates an interesting point about BRCA1 variants.

My main concern with the present manuscript is that as it stands it is very difficult to follow unless you read the authors’ previous manuscript. The reader is constantly referred to a previous paper and the present manuscript lacks some important details. For example what is considered frequent and infrequent variants? It is understandable that the authors do not want to repeat the cohort’s characteristics since it has been published but it would help the reader if a brief summary was provided.

For the benefit of non-specialists, could the authors provide a basis for using penetrance in fertile ages as a marker for fitness? Also, the manuscript doesn’t discuss whether carrying a frequent variant may affect fertility of the carriers which could have implications for the prevalence of these variants.

The paper would also benefit from a thorough revision as there are several awkward sentences, typos and missing axes titles for Figure 2.

Author Response

In this paper Moller et al. examine the penetrance of BRCA1 pathogenic variants that are frequent in the Norwegian population.  Penetrance was assessed in fertile years as a surrogate for evolutionary fitness. They assessed penetrance using an observational prospective study of Norwegian female carriers. These results were compared with penetrance of other not so frequent pathogenic variants in BRCA1 in the Norwegian population and with the average penetrance reported in other breast cancer studies. The authors determine that penetrance of these frequent pathogenic variants was significantly lower than for carriers of infrequent pathogenic variants. They conclude that the relative high prevalence of some pathogenic BRCA1 variants in the Norwegian population is due to genetic drift (although not directly assessed in this paper) and the low selective disadvantage of carriers of frequent pathogenic variants.

This is a short and focused manuscript that reports interesting observation that may however have limited implications for other populations but illustrates an interesting point about BRCA1 variants.

Response: The paper may to us be of interest to explain the frequencies of pathogenic variants in other genes as well: The paper uses BRCA1 as a model, but there is nothing in the methods specific to the BRCA1 variants described.

My main concern with the present manuscript is that as it stands it is very difficult to follow unless you read the authors’ previous manuscript. The reader is constantly referred to a previous paper and the present manuscript lacks some important details. For example what is considered frequent and infrequent variants? It is understandable that the authors do not want to repeat the cohort’s characteristics since it has been published but it would help the reader if a brief summary was provided.

Response: A new section in the introduction summarizing the content of the previous reports is inserted as requested.

For the benefit of non-specialists, could the authors provide a basis for using penetrance in fertile ages as a marker for fitness? Also, the manuscript doesn’t discuss whether carrying a frequent variant may affect fertility of the carriers which could have implications for the prevalence of these variants.

Response: A discussion of this is included in introduction.

The paper would also benefit from a thorough revision as there are several awkward sentences, typos and missing axes titles for Figure 2.

Response: Corrected

Reviewer 2 Report

The authors discussed about the Norwegian population. They try to demonstrate that the frequent deleterious mutations in this population have different impact in the cancer risk. They illustrate the variability on penetrance in the different BRCA1/2 variants. Major questions are the existence of possible biases in this analysis.

The subject is of interest and need to be published as it raises the question at a population level on the recurrent mutations which looks deleterious but which could be less stringent on the risk than the rare mutations.

Some remarks to improve the current article

1- The choice of "path BRCA1" is not a correct wording. It is better to describe as frequent deleterious variant / pathogenic variant. In the introduction, the authors need to describe precisely the name of the frequent deleterious variants.

2- The position on the poly-A tracts is not clear and based on scientific proofs. The authors should position the haplotype in the Norwegian population for thos recurrent deleterious mutations.This should be positionned on the discussion and not in the introduction.

3- One question remains. Does this result connect to the 4 reccurent deleterious variants or to one of them ? A supplementary data could be valuable to explain the impact on each variant on this result. The author need also to provide the list of the mutations included in this study, especially on the non frequent mutations. They also have to validate the absence of bias in those mutations.

4- The data are very convincing however, the author need to reduce the inference on the cause related to those impact. Some discussion can be also proposed on alternative protein related to those mutations which could have a partial impact on the HRD mechanism. This article needs to open on scientific discussion to explain the lower incident - RNA study / tumor study...

5- A discussion on the ashkenaze mutations is aslo needed as some articles have been published also on those mutations. What is the added value of the Norwegian population ? 

6- Finally, some discussion on the management of those mutations is needed, especially in the comment related to a mutation and the hotspot status of it. What about the response to PARP inhibitors ?

Author Response

The authors discussed about the Norwegian population. They try to demonstrate that the frequent deleterious mutations in this population have different impact in the cancer risk. They illustrate the variability on penetrance in the different BRCA1/2 variants. Major questions are the existence of possible biases in this analysis.

The subject is of interest and need to be published as it raises the question at a population level on the recurrent mutations which looks deleterious but which could be less stringent on the risk than the rare mutations.

Some remarks to improve the current article

1-      The choice of "path BRCA1" is not a correct wording. It is better to describe as frequent deleterious variant / pathogenic variant.

Response: ‘pathogenic variant’ is the correct nomenclature according to the BRCAExchange and LOVD/InSiGHT databases for each single variant, and for which there is a need for one short acronym to denote/refer to all pathogenic variants in one gene, which we in the first sentence define to be path_BRCA1 in this report. This to be  consistent with our standard terminology in similar reports (example: PMID: 28754778) and our website referenced by the InSiGHT/LOVD database for path_MMR variants (https://insight-database.org/genes) to consider risk for cancer in specific organs in the various genes path_MMR variants (www.PLSD.eu )

2-      1_b In the introduction, the authors need to describe precisely the name of the frequent deleterious variants.

Response: done.

3-      The position on the poly-A tracts is not clear and based on scientific proofs. The authors should position the haplotype in the Norwegian population for thos recurrent deleterious mutations.This should be positionned on the discussion and not in the introduction.

Response: Description of the poly-A part demonstrated to be subjected to multiple mutations is now included in revised mns and discussion limited to describe this.

 4-      One question remains. Does this result connect to the 4 reccurent deleterious variants or to one of them ? A supplementary data could be valuable to explain the impact on each variant on this result. The author need also to provide the list of the mutations included in this study, especially on the non frequent mutations. They also have to validate the absence of bias in those mutations.

Response: Figures visualizing the cumulative risks for each of the four frequent path_BRCA1 variants described separately is now included. Table with all pathogenic variants described and their frequencies is now also included as on-line supplementary file.

 5-      The data are very convincing however, the author need to reduce the inference on the cause related to those impact. Some discussion can be also proposed on alternative protein related to those mutations which could have a partial impact on the HRD mechanism. This article needs to open on scientific discussion to explain the lower incident - RNA study / tumor study...

 Reponse: We do agree that further studies on why and through which mechanisms the frequent path_BRCA1 variants described have later onset of cancer should be conducted, which are reasons to present the present paper as a platform for why such studies later should be undertaken.

 6-      A discussion on the ashkenaze mutations is aslo needed as some articles have been published also on those mutations. What is the added value of the Norwegian population ? 

 Response: Our topic needs prospective series to be described, and in such you need to eliminate lead-time bias as we have done. We are not aware of any other prospective studies having done so, cfr our discussion of Kuchenbaeker et al claiming (to us not correctly)  their study to be the first prospective study on path_BRCA variants reported, declaring questionnaire-based methods which implies no strict control of lead-time bias.

 7-      Finally, some discussion on the management of those mutations is needed, especially in the comment related to a mutation and the hotspot status of it. What about the response to PARP inhibitors ?

 Response: The topic for our study is age of onset of cancer in path_BRCA1 carriers. Current treatment not available before the current generation is not relevant to our conclusions, breast cancer was (close to) always lethal in former generations, and fertility following survival due to current treatment is outside the scope of this report.